# Green and Sustainable Hot Melt Adhesive (HMA) Based on Polyhydroxyalkanoate (PHA) and Silanized Cellulose Nanofibers (SCNFs)

**DOI:** 10.3390/polym14235284

**Published:** 2022-12-03

**Authors:** Jaemin Jo, So-Yeon Jeong, Junhyeok Lee, Chulhwan Park, Bonwook Koo

**Affiliations:** 1Green and Sustainable Materials R&D Department, Korea Institute of Industrial Technology, 89 Yangdaegiro-gil, Ipjang-myeon, Seobuk-gu, Cheonan-si 31056, Republic of Korea; 2Department of Chemical Engineering, Kwangwoon University, 20 Kwangwoon-ro, Nowon-gu, Seoul 01897, Republic of Korea; 3Department of Polymer Science & Engineering, Sungkyunkwan University, 2066 Seobu-ro, Jangan-gu, Suwon-si 16419, Republic of Korea

**Keywords:** cellulose nanofiber, polyhydroxyalkanoate, biodegradable, silanization, bioplastic, hot melt adhesives

## Abstract

Polyhydroxyalkanoate (PHA), with a long chain length and high poly(4–hydroxybutyric acid) (P4HB) ratio, can be used as a base polymer for eco-friendly and biodegradable adhesives owing to its high elasticity, elongation at break, flexibility, and processability; however, its molecular structures must be adjusted for adhesive applications. In this study, surface-modified cellulose nanofibers (CNFs) were used as a hydrophobic additive for the PHA-based adhesive. For the surface modification of CNFs, double silanization using tetraethyl orthosilicate (TEOS) and methyltrimethoxysilane (MTMS) was performed, and the thermal and structural properties were evaluated. The hydrophobicity of the TEOS- and MTMS-treated CNFs (TMCNFs) was confirmed by FT-IR and water contact angle analysis, with hydrophobic CNFs well dispersed in the PHA. The PHA–CNFs composite was prepared with TMCNFs, and its morphological analysis verified the good dispersion of TMCNFs in the PHA. The tensile strength of the composite was enhanced when 10% TMCNFs were added; however, the viscosity decreased as the TMCNFs acted as a thixotropic agent. Adding TMCNFs to PHA enhanced the flowability and infiltration ability of the PHA–TMCNFs-based adhesive, and an increase in the loss tangent (Tan δ) and adjustment of viscosity without reducing the adhesive strength was also observed. These changes in properties can improve the flowability and dispersibility of the PHA–TMCNFs adhesive on a rough adhesive surface at low stress. Thus, it is expected that double-silanized CNFs effectively improve their interfacial adhesion in PHA and the adhesive properties of the PHA–CNFs composites, which can be utilized for more suitable adhesive applications.

## 1. Introduction

Plastics are durable, inexpensive, mass-productive, and robust materials that are a necessary part of human life [1]. The production of plastics is continuously increasing, and most plastics demand is for packaging materials. Worldwide, almost 367 million tons of plastics were produced in 2020, and post-consumer plastic waste amounts to more than 29 million tons per year. Of all waste plastics, 78% are reusable through energy recovery and recycling, but 23% are still buried in landfills [2].

As described, packaging plastic waste also accounted for the largest proportion of waste plastics, and it has increased steadily due to the current increase in personal hygiene [3]. Therefore, eco-friendly plastic packaging materials are required to reduce the environmental pollution caused by plastic packaging waste. Thus, the essential functions of packaging plastics include packaging as well as respect for the environment, which is the protection and preservation from outside contamination. For these essential functions as packaging materials, packaging plastics are composed of several materials to control suitable properties such as barrier ability, permeability, and hygroscopicity. For example, thermoplastic polyethylene and polypropylene are used for the flexibility of packaging materials, and polyamide is used to ensure mechanical, oxygen barrier, and aroma barrier properties. In addition, polyvinylidene dichloride has been used for its moisture barrier properties, and ethylene vinyl acetate has been used as a sealant and adhesive for multilayer materials [4]. The adhesive is also a material composed of multilayer packaging, although the contents of the adhesive occupy a small number of packaging materials. However, adhesives significantly affect the mechanical properties and recyclability of packaging materials [5].

Adhesives are normally composed of polymers, tackifiers, wax, and antioxidants. Polymers contribute to the mechanical and rheological properties of adhesives. Other ingredients improve or control the adhesive polymer properties, such as viscosity, rheological properties, and flowability, by influencing the polymer structure and interchain interaction. However, the main components of adhesives classified as human carcinogens are toxic when they are decomposed, which hinders the recycling of eco-friendly packaging. Therefore, eco-friendly and biodegradable polymer-based adhesive materials are required as substitutes for phenolic and formaldehyde-based ones [6].

PHA copolymers are composed of poly(3–hydroxybutyrate) (P3HB) and P4HB, and their mechanical properties are determined by the ratio of P3HB to P4HB and the number of carbons in the side chain. A short chain length PHA, which is composed of up to five carbon atoms, is brittle; however, elastomers and rubber-like properties appear when several carbons on the side chain are longer than 15 or when the P4HB content increases [7]. Thus, PHA with a long chain length and high P4HB ratio have high elasticity, elongation at break, flexibility, and processability. However, PHA showed low flowability due to the high viscosity and molecular weight under these condition, and a low tensile strength might be observed [8]. Therefore, the molecular structures of PHA must be adjusted to control their physical properties for adhesive applications.

Cellulose has attracted considerable attention as a sustainable substitute for petroleum-based polymers. In particular, CNFs produced by chemical and mechanical treatment of cellulose, such as oxidation, acid hydrolysis, grinding, and homogenization processes [9], have good mechanical properties owing to their larger aspect ratio and load dispersion in nanostructure form [10,11]. The CNFs suspension has thixotropic properties and low viscosity owing to the breakdown of the nanostructure when shear stress surpasses the threshold and returns to a robust structure through re-alignment during stress relaxation [12]. These properties of CNFs suspensions have been used to control the viscosity of injection solutions and coating materials such as paints and varnishes [4]. Thus, it was expected that the CNFs could be used as green additives to enhance the rheology and mechanical properties of adhesives.

However, the hydrophilic properties of CNFs hinder their uniform dispersion in hydrophobic polymers [13], and various surface modification processes, such as methylation and acetylation, have been applied to overcome the low dispersibility due to their different polarities [14,15]. Silanization was applied for surface modification in this study because of its low toxicity and chemical stability compared with other hydrophobization processes [16,17]. Hydrophobic CNFs were produced according to the double silanization process developed in a previous study, because conventional hydrophobization using only polysiloxane showed morphological problems or insufficient hydrophobicity [18].

This study aimed to develop eco-friendly adhesives using biodegradable polymers (PHA) and sustainable additives (CNFs). Hydrophobic CNFs were prepared using two types of polysiloxanes as additives. Silanization of the CNFs was conducted using TEOS or MTMS. It is well known that the TEOS treatment of cellulose can increase the interaction between the composite phases [13,19], and MTMS, which is an alkoxysilane containing CH_3_ groups, is commonly used to increase the hydrophobicity of cellulose or other OH-group-rich surfaces [20,21]. These multiple steps of silanization using TEOS and MTMS were applied to obtain SCNF additives with dimensional stability and hydrophobicity. The conditions of multi-step silanization, such as reaction time, were optimized, and the optimal condition for compounding between the hydrophobic CNFs and PHA was determined through analysis of the mechanical and rheological properties of the PHA–CNFs composites for the development of eco-friendly adhesives.

## 2. Materials and Methods

### 2.1. Materials

CNFs suspensions were provided by MOORIM P&P (Seoul, Korea). The provided CNFs have a nanoscale needle-shaped structure (Figure 1) with a diameter of less than 30 nm and a length of more than 100 nm. PHA pellets were provided by CJ CheilJedang (Seoul, Korea). TEOS solution (reagent grade, 98%) and MTMS solution (extra pure grade, 96%) were purchased from Sigma-Aldrich (St. Louis, MO, USA) and Daejung Chemicals & Metals (Siheung-si, Korea), respectively. Ammonia solution (extra pure grade, 25–30%) and isopropyl alcohol (IPA, ultrapure grade, >99.9%) were purchased from Samchun Chemicals (Seoul, Korea). The steel plate (114 × 25 mm) used in the lap shear test was purchased from Qmesys (Uiwang-si, Korea).

### 2.2. Hydrophobization of CNFs through Surface Modification

CNFs hydrophobization (Figure 2) was performed following the modified Stöber method [22]. CNFs suspensions (10 mL, 1 wt%) were mixed in 40 mL of 82% IPA, and the CNFs solutions were homogenized for 2 min at 6000 rpm using an ultra-turrax. The alkaline conditions were then adjusted with 1 mL of ammonia solution. To maintain the fibrous structure, the core–shell structure was synthesized through the spherical growth of SiO_2_ on the surface of the CNFs. For the synthesis of the core–shell structure, TEOS of 1 mL was added dropwise, and the mixture was stirred at 500 rpm. TEOS core–shell CNFs (TCNFs) are hydrophilic because they are rich in OH groups. Thus, the OH groups of the TCNFs were shielded using the CH_3_ groups of MTMS to increase hydrophobicity. MTMS (5 mL) was then added dropwise to the TCNFs suspension and stirred at 500 rpm. The TEOS and MTMS reaction times were optimized at 1, 3, 6, and 8 h, and 2, 4, and 8 h, respectively. After finished reaction solutions were centrifuged at 1000 relative centrifugal force (RCF) for 5 min, supernatant was discarded, and residue was washed three times using ethanol to eliminate reaction residue. The TEOS–MTMS-treated CNFs (TMCNFs) were freeze-dried and thermally heat-cured at 150 °C in a vacuum oven.

### 2.3. Characterizations of Modified CNFs

To optimize the TEOS core–shell reaction, such as time and CH_3_ end-capping time, the chemical, thermal, and morphological properties of the CNFs were analyzed depending on the other reaction conditions applied. Fourier-transform infrared spectroscopy (FT-IR, Thermo Nicolet 6700, Thermo Scientific, Waltham, MA, USA) was performed to verify the changes in the functional groups and chemical structure bonding on the CNFs surfaces. FT-IR spectra were recorded in the attenuated total reflection (ATR) mode from 500 to 4000 cm^−1^, and the total scan number was 64 at a resolution of 8 cm^−1^.

The thermal degradation behavior was determined by thermogravimetric analysis (TGA) under a nitrogen atmosphere. All samples were heated from 30 to 800 °C at a rate of 20 °C/min in a platinum pan.

The hydrophobicity of the surface-modified CNFs was confirmed by water contact angle (WCA) measurements using a smart drop (FEMTOBIOMED, Seongnam-si, Korea). Each sample was placed on a glass slide, and 8 μL of a water droplet was dropped for measurements. The WCA was calculated using the baseline set between the water droplet and sample.

The shape and morphology of the CNFs and TMCNFs were observed by field-emission scanning electron microscopy (FE-SEM, JEOL, Akishima, Tokyo, Japan) at an acceleration of 5 kV, an emission current of 10 μA, and a working distance (WD) of approximately 8 mm. The samples were coated with platinum to prevent surface charging effects.

### 2.4. Preparation of PHA–CNFs Composites

PHA–CNFs composites were compounded by hot melt extrusion using a Minimax Molder (Custom Scientific Instruments, Easton, PA, USA). The extruded sample was pelletized and molded using a hot press machine with a size of 100 × 100 mm and a mold of 1 mm thickness. The molding conditions were 170 °C and 5 MPa for 2 min. The molded samples were cut using a JISK 6251 cutter (Figure 3a). Compound ratios of 2%, 5%, and 10% were applied to determine the effect of the ratio on the PHA–CNFs composites. The prepared PHA–CNFs composites are described below.

### 2.5. Characterizations of PHA–CNFs Composites

The dispersibility of the hydrophobized cellulose was verified through FE-SEM analysis of the PHA–CNFs adhesive cross-section. The operating conditions were adjusted to be the same as those for the CNFs morphology analysis. In addition, physical properties such as tensile strength, thermal properties, lap shear test, and rheology behavior were determined to evaluate the feasibility of the PHA–CNFs composite as an adhesive [23,24].

A tensile strength test was carried out to determine the effect of the CNFs compounding ratio on the mechanical properties of the PHA–CNFs adhesive. Tensile test specimens with a width of 10 mm, gauge length of 30 mm, and thickness of 1 mm were prepared in accordance with JISK 6251 (Figure 3a). Tensile strength analysis was performed at a crosshead speed of 500 mm/min using a universal testing machine (UTM, QMESYS, Uiwang-si, Korea) equipped with a 50 kgf load cell, and machine controls and data analysis were performed using the Quality Measurement System, version 12.6.0.

Glass transition temperature (T_g_), melting temperature (T_m_), enthalpy of melting (H_m_), crystallization point (T_c_), and heat of crystallization (H_c_) were measured using differential scanning calorimetry (DSC, PerkinElmer, Waltham, MA, USA) to determine the effect of the CNFs compounding ratio on the crystallization behavior of the PHA–CNFs adhesives. To determine the effect of CNFs on the crystallization behavior of PHA, T_g_, T_m_, H_m_, T_c_, and H_c_ were measured using DSC to confirm the effect of the CNFs compounding ratio on the crystallinity of the PHA–CNFs adhesives. PHA–CNFs adhesives were heated from 30 to 200 °C at a rate of 20 °C/min and held for 5 min to erase the thermal history. Then the samples were cooled from 200 to −50 °C at a rate of 10 °C/min to measure crystallization behavior and the samples were again heated to 200 °C at a rate of 10 °C/min to measure melting behavior, and the machine controls and data analysis were performed using the Pyris-Instrument Managing Software, version 11.

The failure load was analyzed through a lap shear test using a UTM to verify the effect of the tensile strength of the PHA–CNFs composite on the adhesive properties. Lap shear test specimens were prepared by melting the PHA–CNFs adhesive on a steel plate for 10 min at 160 °C, pressing for 5 min at 5 MPa using the other steel plate, and cooling at room temperature. A steel plate was prepared according to ASTM–D1002 for the analysis under the same adhesive thickness conditions. In addition, sufficient samples were produced for the lap shear test, and samples with equal adhesive layers were selected. The test was performed in accordance with ASTM–D1002 (Figure 3b,c).

Changes in rheology properties were analyzed using a rotational rheometer (TA Instruments Ltd., USA) to confirm the effect of CNFs compounding on the flowability of the PHA–CNFs adhesives. Rheological behavior was determined at 160 °C on a cone-plate unit (plate diameter 25 mm, measuring gap 1 mm). The rheology curves presented frequency dependences of storage modulus (G′), loss modulus (G″), and complex viscoelasticity at 0.1–100 Hz, and strain amplitude value of 1%. Tan δ was calculated using the following equation:(1)$$\mathrm{Tan}\;\delta =G^{\prime \prime }/G^{\prime }$$

## 3. Results and Discussion

### 3.1. Characterizations of Modified CNFs

According to the change in mass loss, the TGA curve of the CNFs can be divided into three regions (Figure 4) [25]. In the Section 1 (30–250 °C), both CNFs and TCNFs exhibited mass loss owing to moisture vaporization, and more mass loss was observed in the TGA curve of the TCNFs. TEOS leads to the formation of matrix structures as a precursor of silica core–shell, and the matrix structures facilitate the vaporization of molecular-bound water. However, there was no mass loss in the TGA curve of the TMCNFs because the OH groups on the CNFs surface were reduced by the replacement of terminal OH groups with CH_3_ groups [26]. In the Section 2 (250–350 °C), mass loss of CNFs, TCNFs, and TMCNFs (reaction time < 4 h) was initiated, and it was caused by thermal degradation, such as hydrolysis, oxidation, dehydration, and depolymerization of the glycosidic bond composition [27]. A lower mass loss was observed in the TCNFs and TMCNFs than in the CNFs. Silica core–shell synthesized on the surface of TCNFs and TMCNFs prevented outgassing of thermal degradation products derived from the glycosidic bond, and the remaining thermal degradation products were regarded as the remaining mass. In the case of TMCNFs (reaction time > 4 h), the least mass loss was observed due to the sufficient synthesis of the silica core–shell of the surface [22]. In the Section 3 (350–800 °C), only a small mass loss was observed as the temperature increased. It was assumed that most organic constituents were already degraded in the Section 2, and that the small mass loss was caused by the dehydroxylation of some organic residues [26].

Different mass decomposition curves were observed between the CNFs samples through TGA, and it was confirmed that core–shell synthesis and CH_3_ end capping were successfully performed through silanization using TEOS and MTMS.

The silanization of CNFs using TEOS and MTMS was confirmed by FT-IR as shown in Figure 5. In the spectra in Figure 5a, the peaks derived from the surface groups of the CNFs were observed at 1421 cm^−1^ (C–H symmetrical bending vibration), 1321 cm^−1^ (C–H deformation), 1161 cm^−1^ (C–O–C asymmetric stretching), and 896 cm^−1^ (C–H deformation in glycosidic linkages) [28]. In Figure 5b, these peaks disappear, whereas the peaks corresponding to the silanized surface appear at 800 cm^−1^ (Si–C vibration), 1084 and 1170 cm^−1^ (Si–O–Si stretching). All these peaks were derived from the silica core–shell structure on the TCNFs surface. However, the cellulose intra- and intermolecular structures (1000–1200 cm^−1^) did not change during surface and selective modification using polysiloxane, and these regions of peaks overlap the cellulose intra- and intermolecular structures’ peaks. Therefore, we confirmed the modification by comparing other peaks, such as the significant decrease in the OH elongation peak due to SiO_2_ nanoparticles shielding (2900 and 3300 cm^−1^), and derived peaks from SiO_2_ nanoparticles (760, 800, 950, and 1270 cm^−1^). In particular, the TCNFs were fully covered by SiO_2_ nanoparticles with OH end groups, which can be confirmed by the peak at 950 cm^−1^. As the SiO_2_ nanoparticles shield OH groups of CNFs by TEOS treatment, the OH stretching peaks at 3300 and 2900 cm^−1^ considerably reduce in Figure 5b,c, which means the TEOS treated CNFs should be more hydrophobic and only little water can be adsorbed on the surface. Coating the TEOS-treated CNFs with MTMS shielded the OH end groups of SiO_2_ to CH_3_ groups by double silanization (simultaneous hydrophobization). Thus, the Si–CH_3_ stretching peaks at 760 and 1270 cm^−1^ were pronounced, whereas the one at 1084 cm^−1^ decreased (Figure 5c) [29]. The FT-IR spectra demonstrated that the surface silanization of CNFs via TEOS and/or MTMS treatment was successful, while the actual hydrophobization was evaluated by WCA measurements.

The WCA indicates the surface characteristics of the material, between hydrophilic and hydrophobic. The WCA of the CNFs films varied depending on the functional groups on their surfaces. The lowest WCA was determined for the CNFs film, which was attributed to the abundant hydroxyl groups on its surface. The TCNFs film showed a relatively low WCA (32°), which means that the TCNFs film was still hydrophilic through its OH-group-rich and rough surface due to the capillary effect (Figure 5a’,b’). However, the film prepared with TMCNFs showed the highest WCA of 177° (Figure 5c’), which means that the end groups on the surface were substituted with CH_3_ and its surface characteristics completely changed from hydrophilic to hydrophobic. This indicated that hydrophobization of the CNFs was successfully achieved by simultaneous silanization using TEOS and MTMS, and improvement of dispersibility in PHA was expected.

Figure 6 shows SEM images of freeze-dried CNFs (FD-CNFs) and TMCNFs. A sheet-like form structure was observed in the SEM image of FD-CNFs. As determined, CNFs have rich hydroxyl groups on their surface and various types of binding occur between the hydroxyl groups and CNFs suspension water. CNFs suspension water can be divided into free water, freezing water, and non-freezing water depending on the hydroxyl group bonding type. Though free water and freezing water are not directly linked to the CNFs hydroxyl groups, non-freezing water is bounded directly to other CNFs hydroxyl groups [30]. Thus, CNFs aggregation occurs by an increase in hydroxyl bonding among non-freezing water on CNFs surface and reducing free-water contents in CNFs suspension during the drying process. This phenomenon is called hornification and it arouses a sheet-like formation [31]. However, a sheet-like form structure was not observed in the SEM image of TMCNFs, because of hindering hydroxyl bonds through the shielding of terminal hydroxyl groups by methyl groups. Therefore, the double silanization can keep the CNFs nanostructure during the drying process.

### 3.2. Characterizations of PHA–CNFs Composites

The dispersion of silanized CNFs in PHA was evaluated by observing the cross-sectional area of the adhesive based on the PHA–CNFs composite. In Figure 7a, CNFs particles with diameters greater than 100 μm are observed in the SEM image, which are especially disparate from PHA. It appears that the particles are much larger than the non-treated CNFs in Figure 7a, which means that the non-treated CNFs are not well dispersed into PHA during the compounding process and cause more aggregation. Figure 7a displays voids in the PHA, where CNFs were pulled out because of the abundant OH groups on the hydrophilic surfaces and their low interfacial adhesion [32]. As shown in Figure 7b, the TMCNFs were well dispersed into the PHA, with no observed voids or TMCNFs aggregation at their cross-section. This indicates that the CH_3_ groups end-capped on the surface of the CNFs OH groups using MTMS successfully improved the interfacial adhesion between PHA and TMCNFs.

The mechanical properties of PHA–TMCNFs-based adhesives were analyzed by tensile strength analysis, and the results were statistically processed by multiple comparison analysis using Tukey’s post hoc test [33], as shown in Figure 8. A post hoc test was performed to compare the variance between the PHA and PHA–TMCNF adhesive groups. In a previous study, a drastic decrease in the dispersibility of hydrophobic CNFs into PHA and aggregation of the CNFs at more than 10% of the compounding ratio was observed, and the high limit of the compounding ratio was set at 10% for this study [34].

Based on the analysis of variance (ANOVA) results (Table 1), it was confirmed that the TMCNFs content significantly affected the mechanical properties of the PHA–TMCNFs adhesives. In addition, it was determined that the results of multiple tests were valid as the *F*-value was higher than 4.07 and the *p*-value was less than 0.05 [35]. The tensile strength and elongation at break of the PHA–CNFs adhesive showed similar tendencies with the previous literature [36]. The tensile strength of the PHA–TMCNF adhesive was higher than that of neat PHA as an increase in content of CNFs and the highest tensile strength (0.963 MPa) was obtained at 10% TMCNFs content in the adhesive. However, the elongation at break decreased compared to that of neat PHA, and a slight increase was observed as the TMCNFs content increased in the adhesive. A decrease in elongation at break by 29% was observed compared with that of neat PHA when 2% TMCNFs were added. These changes in mechanical properties mean that the TMCNFs are dispersed well, and the optimization of the compounding process can enhance mechanical properties such as tensile strength, which is considered to be a weakness of biodegradable adhesives. The increase in tensile strength showed a similar tendency compared with the previous study on the enhancement of mechanical properties of polymers using the nanostructure of cellulose filler [32,36,37]. CNFs have been regarded as additives to reinforce biodegradable polymers such as PHA in previous studies [38]. The reinforcement was caused by the high aspect ratio and nano network structure. In addition, the nano network prevented stress concentration in the composites by acting as a load transfer using the structure of fiber–fiber and fiber–polymer matrix bridges [39]. In addition, CNFs have been used as plasticizers and nucleating agents; thus, they can be applied to overcome the poor mechanical properties of PHA owing to their low crystallinity, slow crystallization, and broad size distribution of spherulites. The application of CNFs to PHA facilitates mechanical reinforcement through the growth of small crystals during the crystallization and an increase in the number of small spherulites [40].

The T_g_, T_m_, and H_m_ were measured during the second heating after erasing the thermal history, because production by melt extrusion and storage at room temperature may affect the thermal behavior of the samples. The T_c_ and H_c_ were measured during the first cooling step after erasing the thermal history (Table 2).

T_c_, H_m_, and H_c_ increased with an increase in the compounding ratio of TMCNFs, which can be associated with the increase in crystalline regions and equalization of spherulite size distributions owing to good dispersion in PHA. Consequently, it was confirmed that the intermolecular forces of the PHA–TMCNFs composites increased with the addition of TMCNFs. The CNFs nano-network structure can stabilize the homogeneous crystallization behavior of polymers as nucleating agents by preventing the merging of PHA crystals during recrystallization [41]. Therefore, it was expected that TMCNFs could act as nucleation agents and enhance the tensile strength of the PHA–TMCNFs composite.

The viscoelastic properties of the adhesives were analyzed by rheometric analysis to confirm the effect of TMCNFs compounding on the adhesive flowability at 0.1 to 100 Hz. The frequency range was based on prior studies [42,43], which could also be verified from the absence of denaturation of the adhesive samples under the analysis conditions by confirming the pseudo-plastic behavior in the storage modulus and viscosity graphs. The storage and loss moduli decreased in the low-frequency region, confirming that TMCNFs affected the rheological properties of the adhesive. However, it was observed that a similar level of storage modulus as PHA and the loss modulus increased (Figure 9a,b). Thus, Tan δ affected by loss and storage modulus was maintained above 1 at a high frequency (Figure 9d), which means that the PHA–TMCNFs based adhesive shows a similar level of viscous behavior as neat PHA and it can be utilized an adhesive. In the case of viscosity, lower viscosity was observed than that of neat PHA at all frequency sections (Figure 9c). The viscosity determines the flow and infiltration ability of the adhesive, and the low viscosity of the PHA–TMCNFs suggests the feasibility of a PHA–TMCNFs-based adhesive [44].

When the TMCNFs were applied as the additives for the PHA-based adhesive, it was expected that the TMCNFs would act as a thixotropic agent. The application of TMCNFs decreases the viscosity compared with neat PHA when experiencing shear stress and increases the tensile strength under external stress-relaxing conditions by maintaining a stable nanostructure [12]. An independent samples *t*-test was performed to compare the failure loads of the different TMCNFs content groups. As a result, 1.48 of a *t*-value, and 0.213 of the above 0.05 *p*-value were evaluated. Therefore, the failure load was not statistically significant as the TMCNFs content varied because the null hypothesis cannot be rejected (Table 3) [45]. Based on the results above, it was determined that adding TMCNFs enhances the flowability and infiltration ability of the PHA–TMCNF adhesive. In addition, it was confirmed that an increase in tan δ and adjustment of viscosity by adding TMCNFs without reducing the adhesive strength can improve the flowability and dispersibility on a rough adhesive surface at low stress. Thus, the double silanization of CNFs using different polysiloxanes (TEOS and MTMS) effectively improved the interfacial adhesion in PHA and the adhesive properties of the PHA–CNF composites.

## 4. Conclusions

In this work, we demonstrated a process that enhances the adhesive properties of the PHA-CNFs composite by adding surface-modified CNFs. PHA with a high ratio of P4HB, which is known to have low tensile strength, high elongation at break, and loss modulus, was chosen as the base polymer for biodegradable hot melt adhesive. CNFs were selected as an additive to enhance the characteristics of the PHA-based adhesive, and hydrophobization was performed through silanization to achieve uniform dispersion in the polymer. Double silanization using different polysiloxanes (TEOS and MTMS) was performed, and a well-dispersed PHA–CNFs composite was produced. Several analyses were performed, such as mechanical and thermal properties, to determine the feasibility of the prepared PHA–CNFs composite-based adhesive. In addition, the adhesive properties of the PHA–CNFs composite were evaluated, and it was confirmed that the double-silanized CNFs were effective in enhancing the adhesive properties of the PHA-based hot melt adhesive. Thus, double-silanized CNFs can be considered as a promising material for improving the adhesive properties of PHA-based hot melt adhesives. In addition, this surface modification strategy of CNFs and their compounding in biodegradable polymers shows great potential for the industrial demands of biodegradable polymers.

Finally, adhesive coating conditions, such as thickness, press time, and pressure, will be carried out to improve failure loads, and various applications of the plate, except steel, to expand the utility will be performed in a follow-up study.

## Figures and Tables

**Figure 1 polymers-14-05284-f001:**
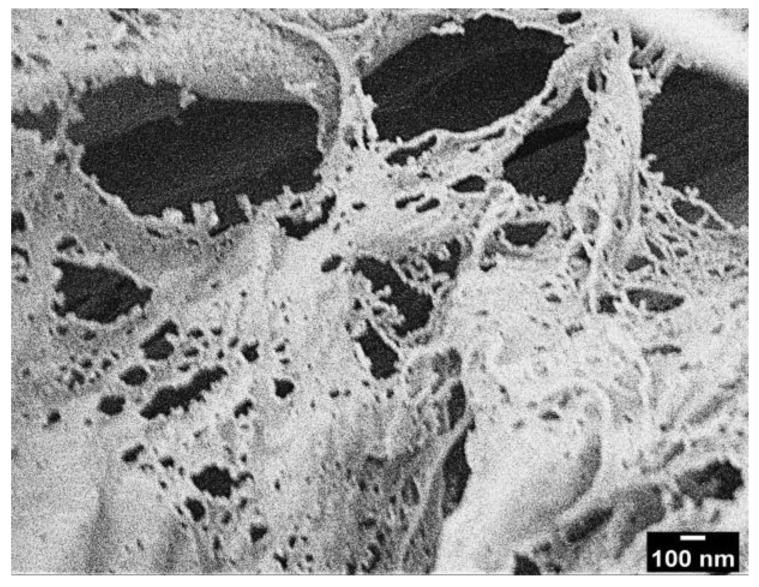
FE-SEM image of FD-CNFs provided by MOORIM P&P.

**Figure 2 polymers-14-05284-f002:**
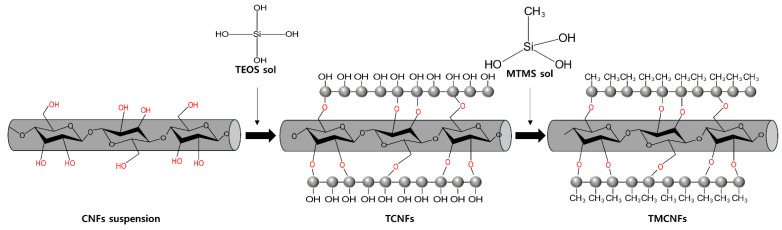
Schematic illustration of surface modification of CNFs using TEOS and MTMS.

**Figure 3 polymers-14-05284-f003:**
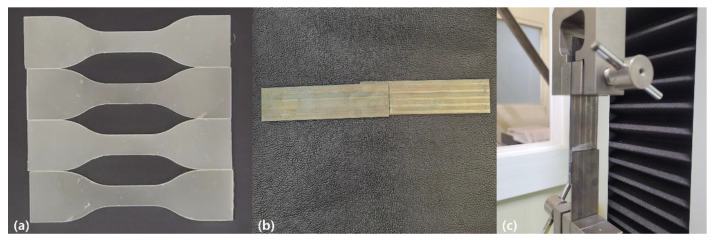
PHA–CNFs composite specimens for (**a**) Tensile test; (**b**,**c**) Lap shear test.

**Figure 4 polymers-14-05284-f004:**
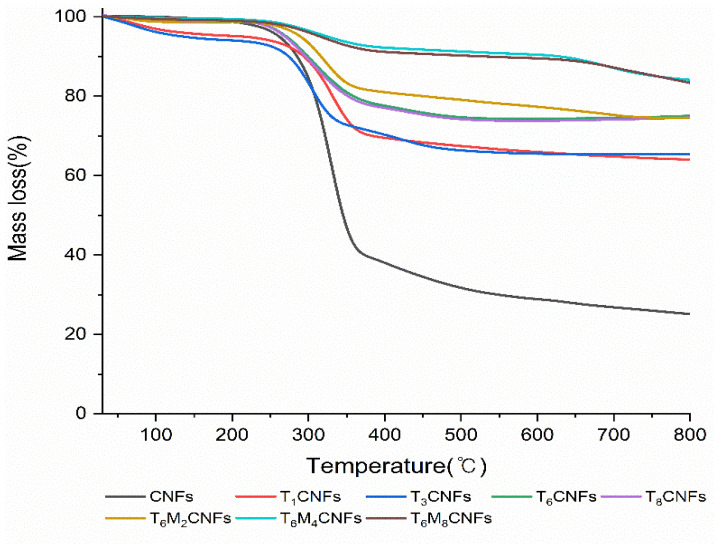
TGA curve of the silica surface modified CNFs (TnMnCNFs means applied reaction times of core–shell synthesis and CH_3_ end capping).

**Figure 5 polymers-14-05284-f005:**
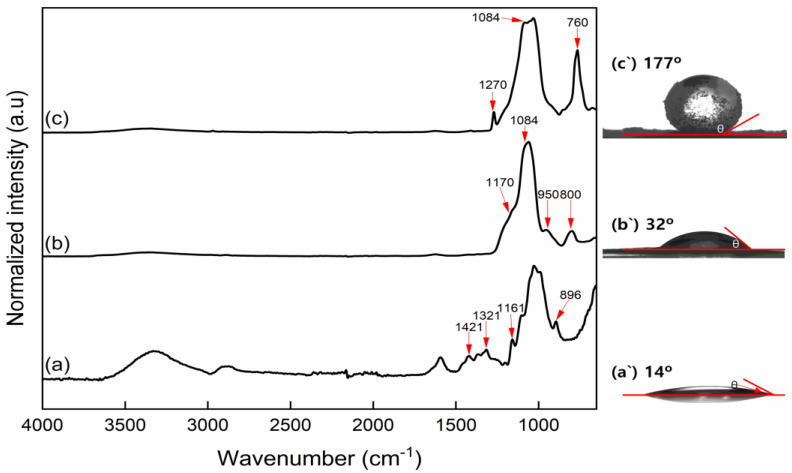
FT-IR peak and water contact angle of the (**a**,**a’**) Non-treat CNFs; (**b**,**b’**) TCNFs; (**c**,**c’**) TMCNFs.

**Figure 6 polymers-14-05284-f006:**
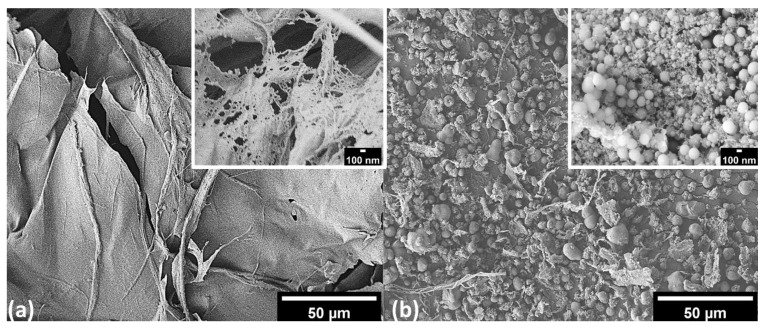
FE-SEM images of (**a**) FD-CNFs; (**b**) TMCNFs.

**Figure 7 polymers-14-05284-f007:**
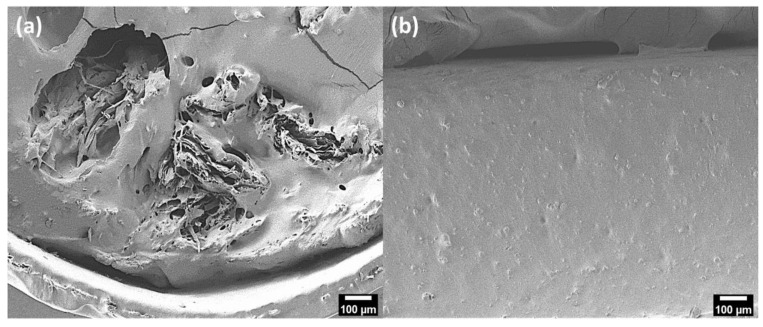
FE-SEM images of cross-section (**a**) FD-CNFs–PHA; (**b**) TMCNFs–PHA.

**Figure 8 polymers-14-05284-f008:**
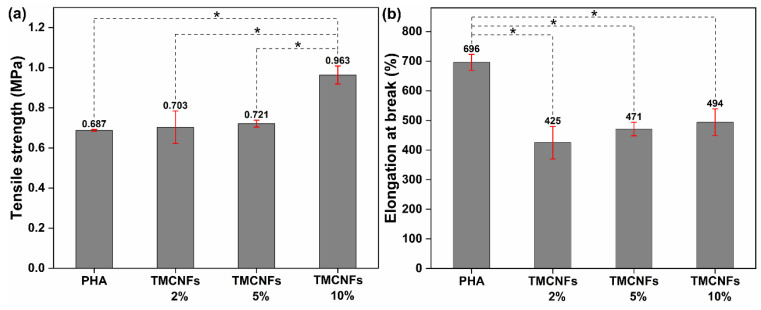
Mechanical properties of the PHA–TMCNFs composites, (**a**) Tensile strength; (**b**) Elongation at break (*: significance level <0.05).

**Figure 9 polymers-14-05284-f009:**
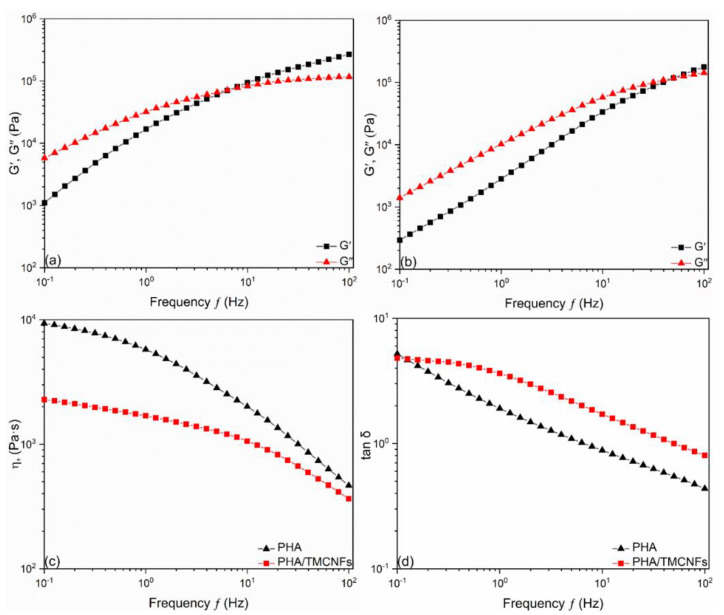
Viscoelastic properties of (**a**) PHA adhesive; (**b**) PHA–TMCNFs 10% adhesive; (**c**) Viscosity of adhesive; (**d**) Tan δ of adhesive.

**Table 1 polymers-14-05284-t001:** ANOVA table of mechanical properties.

Mechanical Properties		Sum of Squares	Degree of Freedom	Mean Square	*F*-Value	*p*-Value
Tensile strength (MPa)	Regression	0.15	3	0.05	7.72	0.01
	Residual	0.05	8	0.01		
	Total	0.20	11			
Elongation at break (%)	Regression	128,997.00	3	42,999.00	9.09	0.01
	Residual	37,838.00	8	4730.00		
	Total	166,835.00	11			

**Table 2 polymers-14-05284-t002:** Summarized thermal properties of the PHA–TMCNFs adhesive.

Sample Name	T_g_ ( °C)	T_m_ ( °C)	H_m_ (J/g)	T_c_ ( °C)	H_c_ (J/g)
PHA	−23	161	0.31	-	-
TMCNFs 2%	−24	159	0.43	75	1.44
TMCNFs 5%	−22	160	0.58	81	1.67
TMCNFs 10%	−23	160	0.69	79	2.15

**Table 3 polymers-14-05284-t003:** Independent samples *t*-test of PHA and PHA–TMCNFs failure load.

Mechanical Properties	Group	N	Mean	Standard Deviation	*t*-Value	*p*-Value
Failure load (kN)	PHA	3	409	41	1.48	0.213
	PHA–TMCNFs 10%	3	340	69		

## Data Availability

The data presented in this study are available on request from the corresponding author.

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
