# Peer review of "Green and Sustainable Hot Melt Adhesive (HMA) Based on Polyhydroxyalkanoate (PHA) and Silanized Cellulose Nanofibers (SCNFs)"

_polymers, 2022, doi:10.3390/polym14235284_

Round 1
Reviewer 1 Report
1. It is important to present a scheme of the hydrophobization reaction. In the same sense, the chemical structures of the materials used.
2. In the topic Characterizations of PHA-CNFs Composites, was there control of the final thickness of the adhesive layer in each application? Since adhesive thickness is an important variable in the adhesion process, this information must be added. Also, how was the thickness of the adhesive layer measured?
3. On the same topic, were the rheological analyses performed within the adhesive's linear viscoelasticity regime? How was this verification done? What was the strain amplitude value used?
4. On page 5, in the text where it says Figure 3 should be Figure 2. The same occurs on pg. 6, where it says Figure 4. And the following Figures are wrongly numbered.
5. From pg. 5, discussion with FT-IR, among the indications of modifications is the observation of new peaks, among them in 1084 and 1170 cm-1 referring to Si-O-Si. However, the literature indicates the region between 1200 - 1000 cm-1 as the occurrence of C-O-C vibrations; an example is the cited Reference [25]. The non-treat CNFs sample already peaked at approx. 1030 cm-1.
Clearly, there was a modification of the surface, but could these peaks not have been superimposed in the region of the cited peaks? Was there any analysis on this?
6. In Table 2, the increase in Hm usually causes an increase in the material's crystallinity. Was this parameter calculated from the DSC data? If yes, are the values significant? In an adhesion process, higher crystallinity may weaken adhesion due to a decrease in the amorphous concentration.
7. Discussion of data should be clearer about which sample is being analyzed and standardize sample nomenclature. The mechanical behavior of the sample is denominated as PHA and in the DSC as T6M4CNFs 0%.
8. It is unclear which sample is being analyzed by rheology as well as by lap shear. Particular attention should be given to samples and discussion by lap shear. What conditions were analyzed, and what was the trend of the results?
Author Response
The authors have tried to revise the manuscript appropriately and to address suitably according to the reviewers’ comments. All changes based on the comments are shown using by track changes function. Also an English proofreading for the revised manuscript has been performed by a qualified editing company. We attach the certificate for the proofreading.
I attached “1 manuscript (doc) including 9 figures (doc) and 3 tables (doc)”, 2 response to reviewers and the certificate for the proofreading.
If you have any questions or need additional information, please contact me electronically.
Thank you for your kind consideration.

Reviewer 2 Report
Comments and Suggestions for Authors
1. In the abstract, the terms CNFs and tan δ are not described like the others.
2. The phrase: "The essential functions of packaging plastics are respect for the environment" must be explained or reformulated for a better understanding
3. “For these essential functions, packaging plastics are composed of several materials”. Mention some of these plastics materials and their function
4. Check the punctuation in the third paragraph of the introduction section
5. “However, PHA showed low flowability” Is this a result of the experimentation in the introduction section?
6. Commercial CNFs size and morphology are not described.
7. Verify that there is no methodology development in the introduction section
8. SEM operational parameters are missing
9. Hot melt extrusion parameters are missing
10. What is the reason for 2,5 and 10% compounding ratios?
11. Complete the tensile strength parameters
12. The numbering of the figures does not correspond to the text
13. Include a graphic explanation of the silanization
14. Figure 3 does not have the same scale in (a) and (b) for proper comparison
15. Figure captions are not uniform
16. There is no text uniformity: Elongation at the brake, elongation at brake, Mpa, MPa,
17. “In order to determine the effect of CNFs on the crystallization behavior in PHA, Tg, Tm, Hm, Tc, and Hc were measured using by DSC” This is methodology in the results section
18. Tg, Tm, Hm, Tc, and Hc are not properly described
19. The number of references is not sufficient for this work.

Author Response

(The authors gave the same response as above.)

Round 2
Reviewer 2 Report
Thank you for addressing the comments. After fixing some minor text errors, I consider the document ready to be published.
Author Response
Dear reviewer
We really appreciate your doing for the manuscript and were trying to revise the manuscript based on your comment. We made three corrections using the Check spelling and grammar function in MS word and all changes based on the comments are shown using by track changes function.
Also, the manuscript was proofread by a qualified English editing company. We attach the certificate for the proofreading
Thank you so much again for your effort.
